# Monitoring the Disulfide Bonds of Folding Isomers of Synthetic CTX A3 Polypeptide Using MS-Based Technology

**DOI:** 10.3390/toxins11010052

**Published:** 2019-01-17

**Authors:** Sheng-Yu Huang, Tin-Yu Wei, Bing-Shin Liu, Min-Han Lin, Sheng-Kuo Chiang, Sung-Fang Chen, Wang-Chou Sung

**Affiliations:** 1Mithra Biotechnology Inc., New Taipei City 221, Taiwan; shengyu.huang@mass-solutions.com.tw; 2Department of Chemistry, National Taiwan Normal University, Taipei 116, Taiwan; a67586danielwei@gmail.com; 3College of Life Sciences, National Tsing Hua University, Hsinchu 30013, Taiwan; bingsin@nhri.org.tw; 4National Health Research Institutes, National Institute of Infectious Diseases and Vaccinology, Miaoli 35053, Taiwan; hamlyn@nhri.org.tw (M.-H.L.); shengkuo@nhri.org.tw (S.-K.C.)

**Keywords:** disulfide bond analysis, folding isomer, cardiotoxin, mass spectrometry

## Abstract

Native disulfide formation is crucial to the process of disulfide-rich protein folding in vitro. As such, analysis of the disulfide bonds can be used to track the process of the folding reaction; however, the diverse structural isomers interfere with characterization due to the non-native disulfide linkages. Previously, a mass spectrometry (MS) based platform coupled with peptide dimethylation and an automatic disulfide bond searching engine demonstrated the potential to screen disulfide-linked peptides for the unambiguous assignment of paired cysteine residues of toxin components in cobra venom. The developed MS-based platform was evaluated to analyze the disulfide bonds of structural isomers during the folding reaction of synthetic cardiotoxin A3 polypeptide (syn-CTX A3), an important medical component in cobra venom. Through application of this work flow, a total of 13 disulfide-linked peptides were repeatedly identified across the folding reaction, and two of them were found to contain cysteine pairings, like those found in native CTX A3. Quantitative analysis of these disulfide-linked peptides showed the occurrence of a progressive disulfide rearrangement that generates a native disulfide bond pattern on syn-CTX A3 folded protein. The formation of these syn-CTX A3 folded protein reaches a steady level in the late stage of the folding reaction. Biophysical and cell-based assays showed that the collected syn-CTX A3 folded protein have a β-sheet secondary structure and cytotoxic activity similar to that of native CTX A3. In addition, the immunization of the syn-CTX A3 folded proteins could induce neutralization antibodies against the cytotoxic activity of native CTX A3. In contrast, these structure activities were poorly observed in the other folded isomers with non-native disulfide bonds. The study highlights the ability of the developed MS platform to assay isomers with heterogeneous disulfide bonds, providing insight into the folding mechanism of the bioactive protein generation.

## 1. Introduction

The linkage of disulfide bonds is one of the post-translational modifications involved in protein structure construction and stabilization [1]. The chemistry of disulfide bonding is an oxidization reaction between the thiol groups of two cysteine residues, which are proposed to influence the thermodynamics of protein by decreasing the conformation entropy [2]. In nature, multiple disulfide bonds commonly occur on membrane and secretary proteins, maintaining their bioactive structures and allowing them to perform their designed functions in the extracellular environment. Typically, most disulfide bonds are built in the periplasm of prokaryotic cells or in the endoplasmic reticulum (ER) of eukaryotic cells, which contain isomerases and cofactors that ensure the correct disulfide bond linkages occur [3,4,5]. Recently, advances in genomic technology have allowed the expression of disulfide-rich proteins in in vitro and in vivo folding systems [6,7,8], but the lack of specific enzymes usually leads to the generation of diverse isomers with non-native disulfide bonds, which impacts the product quality [9,10]. Thus, optimization of the manufacturing conditions is essential. Given that the disulfide bond pattern is highly correlated with the bioactive structure, direct identification of the disulfide bonds of folding intermediates could allow the evaluation of folding conditions as a first step towards manufacturing a bioactive product.

Conventionally, the assignment of disulfide linkage sites can be achieved by X-ray, NMR, and Edman [11,12] technology; however, the demand for sample purity as well as the complex preparation steps involved in these techniques restrict their application to the analysis of the disulfide bonds of folding intermediates, which contain multiple isomers. Recently, mass spectrometry (MS) has emerged as an effective tool for disulfide bond identification by analyzing the in-source fragments of disulfide-linked peptides derived from protein targets with or without partial reduction [13]. However, the screening of the tandem MS spectra of corresponding disulfide-linked peptides for disulfide bond identification can be a time-consuming and labor-intensive task, particularly when analyzing a sample with complex proteins. Previously, an MS-based platform coupling with dimethyl labeling (DM) and the RADAR (rapid assignment of disulfide linkage via a1 ion recognition) search engine was developed to overcome these limitations [14]. The dimethyl-labeled protein digest can natively generate the intense a1 ion during the fragmentation of collision-induced dissociation (CID) in mass spectrometry. By using the N-terminal amino acid identities as a threshold, RADAR can automatically match the a1 ions to the possible sequences, compare the observed *m*/*z*, and then present the disulfide-linked peptides. Without sample fractionation, the developed MS-based platform successfully identifies the disulfide bonds of complicated toxin proteins in cobra venom [15]. Thus, this simple and automatic approach is advantageous for identifying the number of the disulfide bonds in samples with isomer mixtures.

Cardiotoxin A3 (CTX A3), also known as cardiotoxin analogue III, is a 60-residue, three-finger toxin that is commonly found in cobra venom. Structure and functional assays indicated that CTX A3 contains the unique β-strand loop structure that is responsible for lysing the cell membrane, resulting in the development of severe necrosis syndrome in cobra-bite victims [16,17]. Ironically, CTX A3 is also a lead compound for novel drug development due to its great specificity on protein targets within cells [18,19,20]. Today, there is a great interest in manufacturing this native-like cardiotoxin molecule due to its restricted source (snakes) [21,22]. In the study, an in vitro oxidative folding reaction was performed on a synthetic CTX A3 polypeptide (syn-CTX A3) in the presence of redox reagents, and the developed MS-based platform was applied to identify the disulfide bonds in folding isomers. The disulfide mapping results were expected to provide information about the folding conditions that are necessary for the manufacture of a well-folded protein product with biological functionality. 

## 2. Results and Discussion

### 2.1. Preparation of Synthetic Polypeptide for Folding Reaction

Typically, folding reactions are initiated by a fully reduced polypeptide that mimics the initial state of ribosome-translated material and allows natively-interacting amino acids to establish a well-defined structure [21]. This fully-reduced material is normally acquired from the inclusion body of a bioexpression system, while the steps of peptide extraction, reduction, and purification are still tedious. 

In the presented study, a microwave-based solid phase peptide synthesis system (MW–SPPS) was used to generate a 60-residue long polypeptide with eight cysteine residues (Appendix A). Microwave radiation can induce molecule vibration, which improves amino acid conjugation efficacy and reduces chain aggregation along the synthetic process [23,24]. The process was accomplished within hours, and nearly half a gram of synthetic polypeptide with a crude purity over 85% (UV 214 nm, Figure 1A) was obtained in a single run. The molecular weight of the synthetic polypeptide, as determined by the MS analysis, was 6747.40 Dalton (Da), which is 1 Da less than the theoretical mass value of the primary sequence of native CTX A3 (6748.29 Da) due to the C-terminal residue being amidated during the dissociation of the peptide from the resin support. 

In the final step of peptide synthesis, the acidic solvent of trifluoroacetic acid (TFA) was used to cleave the peptide from the resin support and remove the protective group from the side chain of the residue. It is known that the use of a low pH buffer can preserve cysteine sulfhydryl—also known as a thiol group—in the reduced state. Indeed, MS showed that the molecular weight of amidated polypeptide was 7 Da more than the molecular weight of intact CTX A3 toxin (6740.20 Da), which indicates that these eight cysteine residues were spatially separated on the synthetic polypeptide. In contrast to acquiring the reduced peptide from the inclusion body, MW–SPPS could be an alternative and reliable platform for the generation of a large quantity of reduced peptide for the folding application.

### 2.2. Mapping the Disulfide Bonds of Folding Intermediates

After confirmation of the primary sequence, the oxidative folding of syn-CTX A3 polypeptide was performed in the buffer with GSH/GSSG redox reagent to enable the catalysis of thio/disulfide exchange of folding intermediates [25]. The intermediates from different time points of the folding reaction were sampled, digested, and dimethylated for use in the tandem MS (MS/MS) analysis. This novel workflow incorporates trypsin digestion at a low pH buffer (pH 6.5) to avoid possible disulfide scrambling in the analysis runs [26]. After RADAR search with the acquired MS/MS spectra, 13 disulfide-linked peptides were repeatedly identified across the folding reaction, as shown in Table 1. The disulfide bond locations as well as the a1 ions are shown. By comparing with the MS/MS spectra of native CTX A3 digest, two of the disulfide-linked peptides were identified as peptides with the native cysteine pairings [15]. Thus, their MS/MS spectra were manually inspected to validate the identity of the disulfide bond mapping results. 

Figure 2A shows the MS/MS spectrum of the peptide ion of *m*/*z* 565.32 (2+), which was identified by the RADAR search engine to have one native cysteine pairing, denoted 1DS–A3 in the study. Two distinct a1 ions, *m*/*z* 106.06 and 119.11, were observed in the MS/MS spectrum, which indicates that 1DS–A3 is composed of two peptide fragments with N-terminal amino acids of cysteine (Cys, C) and asparagine (Asp, N). Additionally, the y ion series along with the precursor molecular weight, 1128.65 Da, allowed the RADAR algorithm to identify its composing peptide sequences as CNK and NLCYK. Based on the peptide fragments and the *m*/*z* of precursor ion, the disulfide bond of 1DS–A3 was identified to have a cysteine pairing of Cys3–Cys21. By comparing with the MS/MS spectrum of precursor ion *m/z* 565.32 (+2) derived from native CTX A3 digest (Figure 2C), a high similarity was observed on the CID fragments which confirms the identity of cysteine pairing on this disulfide-linked peptide.

Figure 2B presents the MS/MS spectrum of precursor ion *m/z* 701.32 (4+) which identifies the composing peptides as TCPAGK, GCIDVCPK, YVCCNTDR, and CN-amide based on observation of CID fragments and four a1 ions of threonine (*m/z* 106.11), glycine (*m/z* 62.09), tyrosine (*m/z* 168.13), and cysteine (*m/z* 106.06). Based on the RADRA search result, this disulfide-linked peptide contains six cysteine residues that correspond to three cysteine pairings in the fully-oxidized status, although it lacks further information to confirm the exact disulfide linkages. By comparing with the MS/MS spectrum of a known disulfide-linked peptide (*m/z* 701.62, +4) derived from native CTX A3 toxin (Figure 2D), both spectra demonstrate similar CID fragments and the same a1 ions, which assists to confirm the disulfide-linked peptide, denoted as 3DS-A3, has the same cysteine pairings, Cys14-Cys38, Cys42-CysC53, and Cys54-Cys59, like that of native CTX A3 toxin. Table 1 also lists non-native disulfide bonds identified in the folding process. For disulfide-linked peptides that contain two cysteines in total, exact disulfide linkages can be determined. For those that contain more cysteines, the linkages between cysteines in close proximity might not be distinguished due to limited CID fragment information. 

### 2.3. Quantitative Analysis of the Disulfide-Linked Peptides

One of the most important applications of MS is the extraction of the peptide chromatogram from the acquired MS dataset, which allows the peak area of the specific peptide ion to be integrated for quantification. Herein, the liquid chromatography equipped with mass spectrometry (LCMS) system was used to analyze the tryptic digest of folding intermediates to measure the peak area of each disulfide-linked peptide across the folding reaction. By plotting the relative peak area percentages against the folding reaction time, the graph demonstrates increases in the quantity of 1DS–A3 and 3DS–A3 over time, reaching a plateau in the late stage of the folding reaction, as shown in Figure 3A. This increasing trend of the peak area graph reflects the continuous generation of their source protein in the folding reaction as well as the stability of their disulfide bonds in the buffer with redox reagents. This reagent-resistant property is in agreement with the findings of an investigation on the CTX analog III, in which NMR analysis showed that seven of eight cysteine residues are buried in the well-folded beta-sheet secondary structure, retarding solvent penetration and leading to the conformation remaining intact in the buffer with low concentrations of thiol reagent [25,27]. Thus, the increasing peak area of the 1DS–A3 and 3DS–A3 peptides strongly indicates that their source proteins have a native-like structure format that buries these cysteine residues as that of the native CTX A3 toxin. 

In parallel, the same LCMS approach was used to measure the peak area of peptides with non-native disulfide bonds across the folding reaction, as shown in Figure 4B. Although a similar trend to that of native disulfide-linked peptides was shown in the first 24 h, the peak area of non-native peptides was found to decrease in the late stages of the folding reaction, suggesting that the source proteins of these peptides are not rigid enough to make the cysteine pairings inaccessible to the redox reagent in the solution. Most of the linkage sites of the non-native disulfide bonds are connected between the adjacent cysteine—e.g., Cys38–Cys42 or Cys53–Cys54—which leads to the constituted isomers having higher entropy due to the small loop closure. Considering that protein folding is related to the change in entropy, these non-native disulfide linkages play a role in a progressive transformation as the protein folded into its native state. Based on the quantitative analysis on the disulfide-linked peptides, it is rational to hypothesize a native-like syn-CTX A3 folded protein with correct disulfide bonds is generated and its quantity reaches a steady level after 48 h folding reaction. 

### 2.4. Secondary Structue Characterization

In the native state, CTX A3 is cross linked by four disulfide bonds that guides the molecular interactions to form the secondary structure of the double and triple beta strands β-sheet. A further structural characterization was performed to validate the generation of a syn-CTX A3 folded protein with a native disulfide bond pattern in the folding reaction, as indicated by the MS analysis. 

Herein, the folding intermediate with 48 h folding reaction was collected for secondary structure analysis. Considering that the diverse structure isomers would interfere with structural characterization, reverse phase LC was conducted to separate the folding intermediate into fractions (Figure 4A) for subsequent circular dichroism (CD) analysis. Figure 4B compares the CD spectra of each fractionated analyte, and it is clearly visible that Fraction H, collected at the elution time of 26.0–26.6 min, showed two characteristic peaks of a β-sheet structure—the positive peak at 195 nm and a broad negative band at 218 nm. In comparison, the other fractions showed a major, negative peak at around 200 nm in the CD spectrum, which is the signature of the random coil structure. Accordingly, the developed MS platform was used to analyze the disulfide bonds in each fractioned analyst, and both 1DS–A3 and 3DS–A3 were simultaneously detected in Fraction H, while these native disulfide-linked peptides were rare or not observed in the other collected fractions.

To summarize, the CD and MS measurements, a syn-CTX A3 folded protein with a native disulfide bond pattern was fractionated using a reverse phase LC system, which presents the same β-sheet secondary structure as that of the native one. Based on the fact that the peak area of the collected Fraction H is about 10% in the elution chromatogram (Figure 4A), an estimated amount of 40 mg well-folded syn-CTX A3 protein was generated in this folding reaction.

### 2.5. Cytotoxic Activity Characerization

Native CTX A3 is a toxic component of *Naja atra* venom that is implicated in cell membrane destruction, leading to severe tissue necrosis in envenoming victims [28,29], and such activity is dependent on the native structure. After structural characterization, an in vitro cell-based assay was used to analyze the cytotoxic activity of fractionated syn-CTX A3 folded protein and a mixture of non-native isomers. Figure 5A,B present the sigmoid growth curve of cells treated with a folded protein and a non-native folded protein mixture, respectively. The cell viability curve shows that the syn-CTX A3 folded protein exerted a potent cytotoxic activity on HL-60 cells, with a half cell growth inhibition effect (IC50) being observed at a dose of approximately 5.112 μg/mL, which is similar to that of native CTX A3 toxin (4.215 μg/mL, Appendix A). In contrast, the non-native isomer mixture did not exert obvious cytotoxicity on HL-60 cells, requiring a concentration nearly four times higher than that of native CTX A3 toxin (IC50 21.93 μg/mL) to lyse half of the HL-60 cells, as shown in Figure 5B. Basically, the cell-based assay results further indicate that the syn-CTX A3 folded protein contains a bioactive structure that can lyse the cell to the same extent as that of the native molecule, demonstrating that such structural activity is highly associated with the disulfide bond pattern. 

### 2.6. Immunogenecity and Antigenecity of the Folded syn-CTX A3 Protein

In addition to the cytotoxicity, changes in the folding structure were found to impact the immunogenicity and antigenicity of the protein. Typically, B cell epitopes are likely to be located around disulfide bonds. Therefore, proteins comprising distinct disulfide bond patterns present different surface structures to B cells and could elicit antibodies with different specificities. 

To assess the effect of disulfide linkages on protein immunogenicity, six-week-old BALB/c mice (n = 6) were immunized intramuscularly with three doses of fractioned analyte in two-week intervals, and then the mouse serum was collected to individually analyze the level of antibody against the native CTX A3. As shown in Figure 6A, a slightly higher Ab titer was found in the antiserum of mice immunized with the syn-CTX A3 folded protein (6.23 ± 0.31, Log10) compared to that of mice immunized with a non-native isomer mixture (4.64 ± 1.16), showing that the syn-CTX A3 folded protein is more immunogenic. Interestingly, both were able to induce the antibody to recognize the native CTX A3.

Accordingly, the neutralization potency against cytotoxicity induced by the native CTX A3 toxin was evaluated in the pooled mouse serum. The antisera raised by a syn-CTX A3 folded protein (Pc = 0.99 mg/g) was approximately eight times more potent in neutralizing CTX A3 as compared to those raised by the non-native isomer (Pc = 0.12 mg/g), as shown in Figure 6B. These results demonstrate that (1) both the syn-CTX A3 folded protein and the non-native isomer could induce the antibody to recognize the native CTX A3 toxin, but (2) only the syn-CTX A3 folded protein can effectively induce the neutralization antibodies through immunization, which indicates the disulfide bond pattern plays a critical role in the immunogenicity and antigenicity of the disulfide-rich immunogen.

## 3. Conclusions

A well-developed MS-based platform was implemented to analyze the disulfide bonds of the structural isomers during the in vitro folding of the syn-CTX A3 polypeptide. A total of 13 disulfide linked peptides containing different cysteine pairings were repeatedly detected from the folding intermediates. Via quantitative analysis on these disulfide-linked peptides, we profiled the changes in the mapped disulfide bonds across the folding reaction, and the non-native cysteine pairings were revealed to be involved in the routes of progressive rearrangement to generate the folded protein with the native disulfide pattern. Together with the HPLC fractionation and structural characterization results, we showed that a syn-CTX A3 folded protein with native-like structural properties was generated in the folding reaction. It is the first time that a native-like CTX A3 protein has been generated using the synthetic polypeptide in the presence of redox reagent of GSSG/GSH. In light of the fact that most disulfide bonds are evolutionally conserved through a three-finger toxin family in cobra species, we speculate that the developed MS platform could be employed to study protein folding reactions in vitro and to facilitate protein manufacturing for biopharmaceutical applications.

## 4. Materials and Methods

### 4.1. Chemicals and Materials

The fluorenylmethoxycarbonyl–amino acids (Fmoc–AA–OH), Oxyma, were acquired from Merck (Heidelberg, Germany). *N*,*N*′-dimethylformamide (DMF), 1-methyl-2-pirrolidone (NMP), dichloromethane (DCM), *N*,*N*′-diisopropylcarbodiimide (DIC), *N*-ethylmaleimide (NEM), formaldehyde-D2, sodium cyanoborohydride, triethylammonium bicarbonate (TEABC), piperazine, isopropanol, sulfuric acid, Freund, incomplete Freund (IPA), ethanol, acetonitrile, trifluoroacetic acid (TFA), and formic acid (FA) were obtained from Sigma-Aldrich (St. Louis, MO, USA). The medium and antibiotics used for cell culture were purchased from GE Healthcare Life Sciences. The cobra venom (*Naja atra*) was acquired from a snake farm (Tainan, Taiwan), and the CTX A3 was purified from crude venom using the method reported earlier [1]. The synthetic peptide of GSSG and GSH was obtained from an NHRI core facility with a purity over 95% (HPLC, UV 214 nm).

### 4.2. Synthesis of the CTX A3 Polypeptide

The synthetic polypeptide sequence was based on the CTX A3 dataset (Accession: P60301) from the gene bank (http://www.ncbi.nle.nih.gov), but the signal fragment from the N-terminal sequence was eliminated, as shown in Appendix A. The synthetic procedure followed the general protocol of the solid phase peptide synthesis (SPPS) method using an automated peptide synthesizer, model Liberty BLUE (CEM, Matthews, NC, USA) coupling with a single-mode microwave reactor. An amount of 0.1 mM of PAL-NovaPEG resin (0.17 mmol/g, Novabiochem®, Merck, Darmstadt, Germany) was used as solid support to attach the amino acids by following the FMOC chemistry protocol [30]. During the synthetic process, a double coupling step was carried out with a mixture of an equal amount of Fmoc-AA-OH/DIC/Oxyma at 90 °C for 125 s. After amino acid conjugation, the Fmoc was de-blocked from the residues with a mixture of piperazine/oxyma in NMP/ethanol (9:1, 4 L) for 65 s at 90 °C. Subsequently, the peptide chain was cleaved from the resin using a solution of TFA/H2O/TIS/EDT (volume ratio 92.5/2.5/2.5/2.5) for 3 h at room temperature. Following that, the crude peptide was precipitated with cold diethyl ether, lyophilized, and then stored at −20 °C until further analysis.

### 4.3. Characterization of Synthetic Polypeptide with LC–UV/MS

The synthetic CTX A3 polypeptide (syn-CTX A3) was characterized using Agilent 1100 series LC/MSD high performance ion trap mass spectrometer coupled with an ultraviolet (UV) detector. A 5 μL sample volume was injected and separated by reverse phase C18 column (Biobasic-18, 150 × 1 mm, 5 μm, Thermo Fisher Scientific, Bellefonte, PA, USA) with a linear gradient from 2% to 85% of mobile acetonitrile (0.1% in FA) at a flow rate of 0.2 mL/min for 40 min. The purity was ensured to be higher than 70% by an optical UV detector with a wavelength at 214 nm. The survey scan ranged from *m*/*z* 300 to 1600, and the acquired MS spectrum was deconvoluted to determine the molecule weight of the peptide analytes.

### 4.4. Oxidative Folding of Synthetic Polypeptide

The synthetic polypeptide (25 mg/mL) was dissolved in the PBS buffer containing 3 mM reduced glutathione, 1 mM oxidized glutathione, 150 mM sodium chloride, 1 mM EDTA, 50 mM arginine, and 50 mM glutamine (pH 7.2). Folding was performed at 4 °C, and the folding intermediates were sampled in a time-course manner. The sampling intermediates were immediately mixed with 1% TFA to stop the folding reaction, lyophilized, and then stored at −80 °C until subsequent analysis. 

### 4.5. Disulfide Bond Analysis

Disulfide bond identification followed the protocol developed by Huang et al. [15]. Briefly, an adequate amount of lyophilized folding intermediate was resuspended with 100 mM TEABC buffer (pH 7.0) and then directly alkylated with NEM without treatment with a reducing agent. Following that, the NEM-alkylated sample was mixed with trypsin (Promega, Madison, WI, USA) at a weight ratio of 1:20 (sample: enzyme) at 37 °C overnight. Subsequently, dimethyl labeling was performed through the addition of 4 μL of 4% (*w*/*v*) formaldehyde-D2 into 160 μL of tryptic digest, followed by 4 μL of 600 mM sodium cyanoborohydride at room temperature for 30 min. The DM-labeled peptide mixtures were ready for the LC–MS/MS analysis. Special caution was taken when handling formaldehyde and sodium cyanoborohydride, including the use of gloves and a fume hood.

An ESI–Q–TOF mass spectrometer Synapt HDMS connected with the nanoACQUITY UPLC system (Waters, St. Milford, MA, USA) was conducted to identify peptide ions in the sample digest. A 5 μL sample of dimethylated digest was injected and then separated with a C18 column (75 μm id × 10 mm, 1.7 μm bead size, Waters) using a linear gradient from 1% to 50% acetonitrile (ACN, 0.1% FA) at a flow rate of 0.2 μL/min for 30 min. The survey scan was from *m*/*z* 400 to 1600, and the MS/MS scan was from *m*/*z* 50 to 1990. The threshold to switch from MS to MS/MS was 40 counts and switch back until the signal was below 10 counts or after 2.4 s. For the DS bond analysis, Masslynx 4.0 Global ProteinLynx was applied to transfer the raw data of MS/MS spectra to the peak list (PKL) for RADAR (http://www.mass-solutions.com.tw/, Mithra, Taiwan) search. The RADAR algorithm was used to screen a1 ions and search for the corresponding molecular weight for DS bond assignment. The mass tolerance was set within ±10 ppm, the a1 tolerance was ±0.002 Da. A dimethyl labeled, intensity ratio cutoff of 10% and a maximum chain number of four were selected for the RADAR search. Up to two missed cleavages were allowed.

For quantitative analysis, the same nanoUPLC–MS analysis conditions were used to analyze the dimethylated digest of the folding intermediates sampled, but the final step of MS/MS sequencing and database searching was eliminated. Each peptide chromatogram was extracted from the acquired MS data by its specific *m*/*z* value (Table 1), and then we proceeded with peak integration to measure the peak area of disulfide-linked peptide. In total, we obtained the peak areas for each disulfide-linked peptide at the folding reaction times of 2, 4, 6, 9 14, 48, 78, 96, and 120 h. For each disulfide-linked peptide, the acquired peak area was normalized to the highest peak area value detected, and such a relative peak area percentage (% peak area) was used to create a time course graph over time.

### 4.6. Fraction of Folding Isomers Using Reverse Phase HPLC

The fractionation of folding intermediates was performed in a high-performance liquid chromatography (HPLC) system (Alliance 2695, Waters, MA, USA) equipped with a dual absorbance ultraviolet detector (Model 2487, Waters, MA, USA). First of all, the sampling intermediates were acidified with TFA (final conc. 0.1%, *v*/*v*) to stop the possible folding reaction, and then they were subjected to a reverse-phase column (Jupiter C18, 250 × 4.6 mm, 5 μm particles with 300 Å pore size, Phenomenex, Torrance, CA, USA) and eluted at 0.8 mL/min flow rate with two different mobile phases (mobile phase B: 0.1% TFA, mobile phase C: 100% ACN/0.1% TFA) at the following gradient: 2% C for 5 min, 2–10% C for 2 min, 10–16% B for 6 min, 16–28% B for 2 min, 28–65% B for 37 min, 65–80% B for 3 min, and 2% C for 10 min. The absorbance of the eluate was monitored at 214 nm. The fraction collection was initiated at an elution time of 21.3 min with an interval of 0.7 min. The collected fractions were lyophilized and then stored at −20 °C until further analysis.

### 4.7. Secondary Structure Analysis Using CD Spectrometry

The lyophilized fractionation was resuspended in a 10 mM phosphate buffer (pH 7.4) at a concentration of 100 μg/mL for CD analysis. The spectrum, which contained an average of five repeats, was acquired on a Jasco J-815 spectropolarimeter (Hachioji-Shi, Tokyo, Japan) at a continuous mode at 20 °C with a 0.2 nm wavelength interval and an accumulation time of 10–15 s/min. The scanning wavelength ranged from 260 nm to 190 nm with a scan speed of 100 nm/min. PBS buffer (10 mM) was applied as a blank control for background subtraction. The secondary structure of the analyte was calculated based on the mean residue ellipticity (θ) at 195 and 214 nm using the algorithm of the Greenfield and Fasman method [31].

### 4.8. Cytotoxic Activity Analysis

The cytotoxicity analysis followed the procedure of a previous study [32]. Initially, 1 × 10^5^ human promyeloblast cells (HL-60, ATCC) suspended in 100 µL of RPMI-1640 complete medium (Hyclone, GE, Logan, UT, USA) and supplemented with 10% heat-inactivated fetal bovine serum and 1% penicillin/streptomycin antibiotics were seeded in 96-well microplates (Corning^®^, Tewksbury, MA, USA). Cells were cultured in a humidified atmosphere at 37 °C with 5% CO_2_ and then treated with various concentrations of folding sample tested (0.625–100 μg/mL). After four hours, 10 μL of CCK-8 solution (Cell Counting Kit-8, Dojindo Molecular Technologies, Inc. Tewksbury, MA, USA) was added into each well, and the plate was further incubated for 4 h for the measurement of cell viability by determining the occurrence of dehydrogenase activity in cells. The absorbance at 450 nm was measured with a microplate reader (SUNRISE, TECAN, AG, Switzerland) and the viability of the untreated cells and medium alone were taken as 100% and 0%, respectively. All treatments were performed in triplicate, and the results were fitted to a dose-dependent sigmoid curve with a variable slope using GraphPad Prism software (GraphPad, La Jolla, CA, USA). Cytotoxicity was expressed as the IC50 value (mM) corresponding to the concentration of toxin, which caused 50% of cell death.

### 4.9. Antigen Immunization

Immunization followed the protocol described in a previous study [33]. In brief, the fractionated sample in PBS buffer was detoxified with glutaraldehyde (0.25% *v*/*v*) at 4 °C overnight. Groups of six mice were primed intramuscularly (i.m.) with 50 µg of detoxified antigen formulated in complete Freund’s adjuvant. Continuous boosting was done twice with the same amount of antigen that was formulated in incomplete Freund’s adjuvant at two-week intervals. Blood was collected by tail vein bleeding two weeks after the final immunization. The acquired serum was decomplemented (30 min at 56 °C) and then stored at −20 °C until use.

### 4.10. Antibody Titer of Mouse Serum against Native CTX A3 Toxin

Serial-diluted antisera (100 µL in PBS/1% BSA, pH 7.5) solutions were added to wells coated with 0.5 µg of CTX A3 and incubated at room temperature for 2 h. The plates were then washed four times with PBST and incubated with goat anti-mouse IgG (1:5000; GeneTex Inc., Irvine, CA, USA) conjugated with HRP for 1 h at room temperature. The assay was developed with TMB for 20 min, then stopped by adding 50 µL of sulfuric acid solution. The absorbance at 450 nm was measured by a microplate reader, and ELISA endpoint titers were obtained from the titration curve by interpolation and defined as the reciprocal of serum dilution that maintained an OD450 value of 0.3. If the OD450 value was <0.3 at the starting dilution or >0.3 at the final dilution, titers were obtained by extrapolation.

### 4.11. In Vitro Neutralization Assay against Cytotoxicity

The neutralization potency evaluation followed the protocol described in a previous study [32]. Different amounts of native CTX A3 (20, 16, 12, 8, 4, 2, 1, 0.5, 0.25, 0.125, and 0.0625 µg/mL) were pre- incubated with 100 µg of pooled antisera in 100 µL of complete medium at 37 °C for 30 min. The antisera–toxin mixture was then applied to 100 µL of 1 × 10^5^ of HL-60 cells in the wells and the viability of cells was determined by the assay described above after 8 h of co-culturing. The efficacy (Pc) of antisera was estimated by the following equation:Pc = V × [IC50(a1) − IC50(a0)]/(a1 − a0)(1)where V is the volume of the well, a is the amount (µg) of antisera, and IC(an) is the IC50 value of CTX A3 neutralized by an amount of antisera.

## Figures and Tables

**Figure 1 toxins-11-00052-f001:**
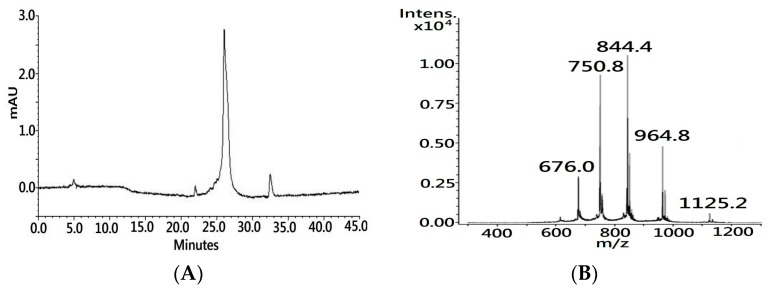
Characterization of synthetic cardiotoxin polypeptide (syn-CTX A3) using liquid chromatography coupled with the ultraviolet detector and electrospray ionization mass spectrometry (LC–UV/ESI–MS). (**A**) One major peak was observed in the LC chromatogram of synthetic polypeptide under the detection wavelength of UV 214 nm. (**B**) The ESI–MS spectrum of synthetic polypeptide showed multiple peak profiles with different charge states: *m*/*z* 676.0 (+10), *m*/*z* 750.8 (+9), *m*/*z* 844.4 (+8), *m*/*z* 964.8 (+7), and *m*/*z* 1125.2 (+6).

**Figure 2 toxins-11-00052-f002:**
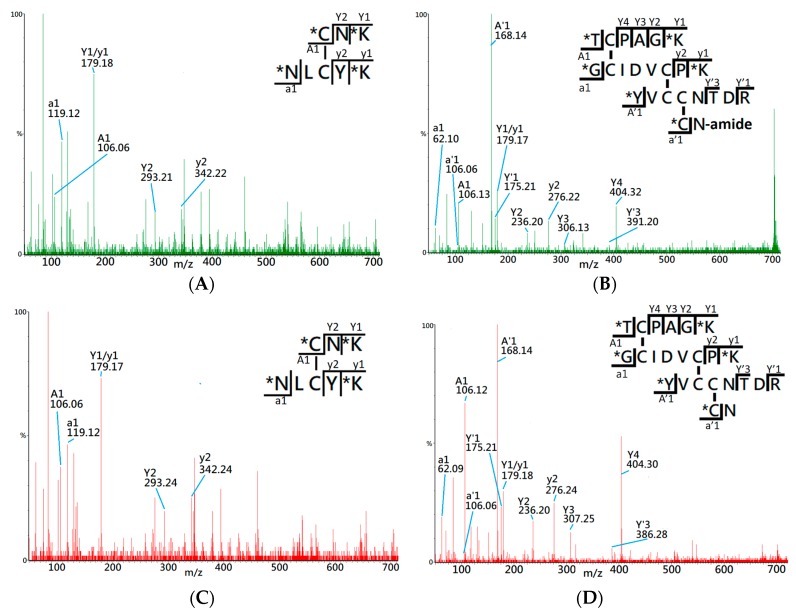
Comparison of MS/MS spectrum of disulfide linked peptides derived from the folding intermediates (**A**,**B**) and native CTX A3 protein (**C**,**D**). The MS/MS spectra were acquired from the precursor ion of (A) *m/z* 565.32 (+2), (B) *m/z* 701.32 (+4), (C) *m/z* 565.32 (+2), and (D) *m/z* 701.62 (+4), respectively. Asterisks (*) indicate the dimethyl labeling sites in the peptides. The CID fragments of the a1/y ion signals are annotated in each spectrum. N-amide represents the amidated asparagine (N) residues in the C-terminals of the peptides.

**Figure 3 toxins-11-00052-f003:**
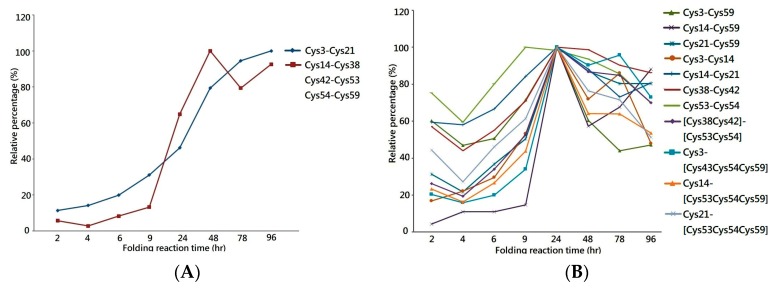
Time course relative peak area graph of (**A**) disulfide-linked peptides with either single or triple native cysteine pairings, and (**B**) disulfide-linked peptides with non-native disulfide bonds. Each relative peak area graph of disulfide-linked peptide was annotated with their corresponding cysteine pairings listed in Table 1.

**Figure 4 toxins-11-00052-f004:**
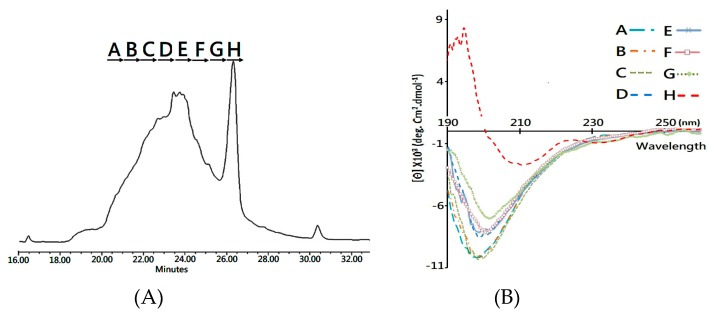
Analysis of secondary structural analysis of fractionated folding intermediate using CD specrometer. (**A**) HPLC chromatogram of folding intermediate (48hr). The sample was separated with reverse phase column, and eight fractions, A-H, were collected for further CD analysis. (**B**) CD measurement of HPLC fraction A to H. The acquired spectra were characterized and combined for comparative analysis.

**Figure 5 toxins-11-00052-f005:**
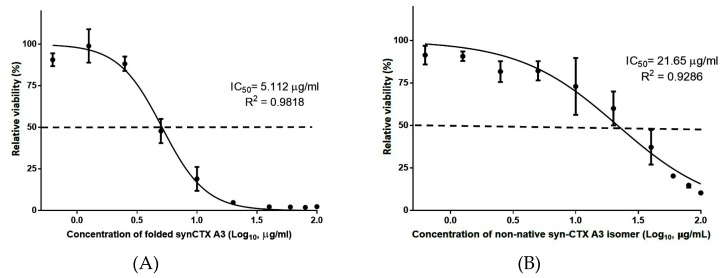
In vitro cell-based cytotoxicity analysis. The sigmoid curve of cell viability was acquired by treating a constant amount of HL-60 cells with incubated with various concentrations of (**A**) folded syn-CTXA3 with native disulfide bonds, or (**B**) folding isomers with non-native disulfide bonds. The assay was performed in triplicate. The dashed line indicates the sample concentration that caused death of half of the cells. The dashed line indicates the toxin concentration that caused death of half of the cells.

**Figure 6 toxins-11-00052-f006:**
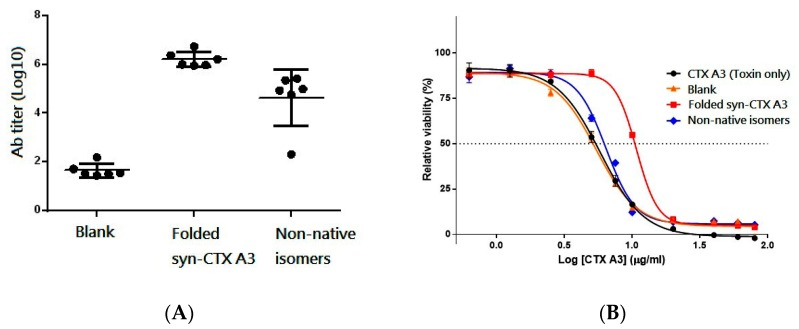
(**A**) The Ab titer (log value) of mouse serum against the native CTX A3 toxin. Blank represents the pre-immune mouse serum for comparative purposes. (**B**) In-vitro cell-based neutralization assay. The sigmoid curve of cell viability was obtained by treating HL-60 cells with a mixture of varying doses of native CTX A3 toxin together with a constant amount of antisera. The cells incubated with native CTX A3 toxin alone were utilized as controls (•) in the assay. The neutralizing potency of mouse antisera against cytotoxicity was calculated by the equation described in the experimental section.

**Table 1 toxins-11-00052-t001:** List of the identified disulfide-linked peptides and the mapped disulfide bonds

Index No.	Disulfide-linked Peptide Sequence ^b^	Cysteine Pairing	a1 ion	Observed *m/z*	Molecule Mass (Da)
1	*C_3_N*K,	**Cys3-Cys21 ^a^**	C 106.06	565.32 (2+)	1128.65
	*NLC_21_Y*K		N 115.08		
2	*TC_14_PAG*K,	**Cys14-Cys38 ^a^**	T 106.12	701.32 (4+)	2801.26
	*GC_38_IDVC_42_P*K	**Cys42-Cys53 ^a^**	G 62.09		
	*YVC_53_C_54_NTDR C_59_(N-amide)	**Cys54-Cys59 ^a^**	Y 168.13 C 106.06		
3	*L*KC_3_N*K,	Cys3-Cys59	L 118.15	483.30 (2+)	964.67
	*C_59_(N-amide)		C 106.06		
4	*TC_14_PAG*K,	Cys14-Cys59	T 106.12	452.74 (2+)	903.55
	*C_59_(N-amide)		C 106.06		
5	*NLC_21_Y*K,	Cys21-Cys59	N 115.08	484.73 (2+)	967.58
	*C_59_(N-amide)		C 106.06		
6	*C_3_N*K,	Cys3-Cys14	C 106.06	533.27 (2+)	1064.54
	*TC_14_PAG*K		T 106.12		
7	*TC_14_PAG*K,	Cys14-Cys21	T 106.12	447.87 (3+)	1340.80
	*NLC_21_Y*K		N 115.08		
8	*GC_38_IDVC_42_P*K	Cys38-Cys42	G 62.09	448.72 (2+)	895.447
9	*YVC_53_C_54_NTDR	Cys53-Cys54	Y 168.13	502.25 (2+)	1002.44
10	*GC_38_IDVC_42_P*K,	[Cys38Cys42]-	G 62.09	633.59 (3+)	1897.93
	*YVC_53_C_54_NTDR	[Cys53Cys54]^C^	Y 168.13		
11	*L*KC_3_N*K,	Cys3-	L 118.15	480.21 (4+)	1917.04
	*YVC_53_C_54_NTDRC_59_(N-amide)	[Cys53Cys54Cys59]^C^	Y 168.13		
12	*TC_14_PAG*K,	Cys14-	T 106.12	465.02 (4+)	1855.92
	*YVC_53_C_54_NTDRC_59_(N-amide)	[Cys53Cys54Cys59]^C^	Y 168.13		
13	*NLC^21^Y*K,	Cys21-	N 115.08	641.02 (3+)	1919.95
	*YVC_53_C_54_NTDRC_59_(N-amide)	[Cys53Cys54Cys59]^C^	Y 168.13		

Note: The bolded cysteine pair represents the native disulfide linkage as found in the native CTX A3 toxin. ^a^ N-amide indicates the amidated C-terminal asparagine. * represents the dimethyl labeling site of peptide with formaldehyde-D2. ^b^ [] are the possible cysteines to be inter-linked with the cysteine on the other peptide chain.

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
