# Peer review of "Monitoring the Disulfide Bonds of Folding Isomers of Synthetic CTX A3 Polypeptide Using MS-Based Technology"

_toxins, 2019, doi:10.3390/toxins11010052_

Round 1
Reviewer 1 Report
A good example of how dimethyl labelling coupled with an MS-assignment software (RADAR) can be used to make the analysis of disulfide connectivity of venom peptides easier than conventional approaches. A few minor things to address, listed below:
Why was the C-terminal acid not made (native peptide)?
Spelling errors: Line 75 ‘Synthetix’ to ‘Synthetic,’ Line 105 ‘Diskfide Bonbds’ to ‘Disulfide Bonds,’ Line 169 ‘nan-native’ to ‘non-native,’ Line 177 ‘rationale’ to ‘rational’
Figure legend for S1: asparagine instead of arginine
Line 82 refers to Table 1, but this doesn’t fit with what the sentence is describing.
For all figures, especially Figs 1-3, the numbers and labels are very small, please make bigger so they are readable.
Line 106: ‘After identification of the primary sequence..’ Reword- it reads as though the peptide was discovered in this paper.
In Table 1, it would be good if the two native disulfide connectivities could be highlighted somehow so readers can easily identify them.
It is also unclear as to what timepoint the data in Table 1 is taken from
Line 130 -131 ‘..four peptide fragments’ however only 3 are in Table 1. Put the C-terminal ‘CN’ on a different row in the table.
Line 133-134 ‘..comparing the MS/MS spectra with that of the native..’ It would be really helpful if the synthetic oxidised MS/MS spectra was overlaid with the native in different colours and included as a figure.
For Figure 3, the relative percentages of each disulfide-linked digested peptides is rather low- none of them are higher than 1% at any time, so at maximum the peptides account for only 13% of the total area- what is the rest? Reduced peptides?
The HPLC trace in Fig S3 is rather informative and could be placed as a part of Fig 4.
For Fig S1 it would be good if lines could be placed in the sequence where trypsin cuts, so readers can more easily see the fragments that would be made.
Author Response
We thank the reviewers for their helpful feedback. Please see our response to reviewer’s comments below.
Reviewer 1.
Comments and Suggestions for Authors
A good example of how dimethyl labelling coupled with an MS-assignment software (RADAR) can be used to make the analysis of disulfide connectivity of venom peptides easier than conventional approaches. A few minor things to address, listed below:
Why was the C-terminal acid not made (native peptide)?
Response: In the study, we used the PAL-NovaPEG resin as solid support to conjugate the amino acids, which is commonly used for synthesizing the long peptide because the PEG chain can increase the hydrophilicity and avoid peptide aggregation during the synthetic process. As the structure shown in Figure R1, PAL-NovaPEG resin consists of aminomethyl-dimethoxyphenoxyvaleric acid linker [Int. J. Peptide Protein Res., 1987, 30, 206] attached to NovaPEG amino resin. The amino group of this linker can be easily acylated under standard coupling conditions. Following peptide assembly, treatment with 95% TFA containing scavengers releases the desired peptide amide.
Figure R1. The structure of PAL-NovaPEG resin.
Spelling errors: Line 75 ‘Synthetix’ to ‘Synthetic,’ Line 105 ‘Diskfide Bonbds’ to ‘Disulfide Bonds,’ Line 169 ‘nan-native’ to ‘non-native,’ Line 177 ‘rationale’ to ‘rational’ .Figure legend for S1: asparagine instead of arginine
Response: Done by correcting these mistyping errors. The words/sentences changed in the content are marked in red.
Line 82 refers to Table 1, but this doesn’t fit with what the sentence is describing.
Response: Thank you for the indication. The (Table 1) was changed to (Figure S1) in the revised manuscript.
For all figures, especially Figs 1-3, the numbers and labels are very small, please make bigger so they are readable.
Response: The labels in the figures are modified as requested in the revised manuscript.
Line 106: ‘After identification of the primary sequence..’ Reword- it reads as though the peptide was discovered in this paper.
Response: The sentence has been changed to ‘After confirmation of the primary sequence’.
In Table 1, it would be good if the two native disulfide connectivities could be highlighted somehow so readers can easily identify them.
Response: a note was added in the Table to indicate the native disulfide linkages.
It is also unclear as to what timepoint the data in Table 1 is taken from
Response: Table 1 listed the disulfide linked peptides that were repeatedly identified across the folding reaction, but not single time point. The description of MS identified peptides is shown in Line 110-111.
Line 130 -131 ‘..four peptide fragments’ however only 3 are in Table 1. Put the C-terminal ‘CN’ on a different row in the table.
Response: Thank you for the indication. The C-terminal CN was put on a different row in the item 2 of table 1.
Line 133-134 ‘..comparing the MS/MS spectra with that of the native..’ It would be really helpful if the synthetic oxidised MS/MS spectra was overlaid with the native in different colours and included as a figure.
Response: Done by adding the MSMS spectrum of disulfide linked peptides derived from native CTX A3 in the figure 2 for the comparative purpose.
For Figure 3, the relative percentages of each disulfide-linked digested peptides is rather low- none of them are higher than 1% at any time, so at maximum the peptides account for only 13% of the total area- what is the rest? Reduced peptides?
Response: thank you for the indication. the y-axis of scale has been changed to the 0-100% to show the changes of amount of each disulfide linked peptide across the folding reaction.
The HPLC trace in Fig S3 is rather informative and could be placed as a part of Fig 4.
Response: Done by adding the HPLC chromatogram in the Figure 4 in the revised manuscript.
For Fig S1 it would be good if lines could be placed in the sequence where trypsin cuts, so readers can more easily see the fragments that would be made.
Response: Done by adding the bottom lines in the peptide chains linked with disulfide bonds.

Reviewer 2 Report
Dear Authors, Overall manuscript is appropriate. Please discuss whether the HPLC method used was validated and how. In addition, please discuss whether FTIR can also be used in addition to the proposed methods?Author Response
We thank the reviewers for their helpful feedback. Please see our response to reviewer’s comments below.
Reviewer 2.
Dear Authors, Overall manuscript is appropriate. Please discuss whether the HPLC method used was validated and how.
Response: In general, we injected the synthetic peptides with different hydrophobicity into HPLC and analyzed with designed elution gradient. Based on t elution time and peak area of each chromatographic peaks, we can validate the the HPLC performance based on each sample analysis.
In addition, please discuss whether FTIR can also be used in addition to the proposed methods?
Response: FTIR is one of optical instrument used for characterizing the secondary structure of proteins based on the detection of amide bands. In general, amide band at 1640 and 1650 cm-1 stands for the secondary structure of beta-strand and alpha-helix, respectively. However, FTIR is not able to detect the analysis the disulfide bonds in samples since the C-S and S-S stretching is not visible in FTIR spectrum.
